# Alumina Manufactured by Fused Filament Fabrication: A Comprehensive Study of Mechanical Properties and Porosity

**DOI:** 10.3390/polym14050991

**Published:** 2022-02-28

**Authors:** Veronika Truxová, Jiří Šafka, Jiří Sobotka, Jakub Macháček, Michal Ackermann

**Affiliations:** 1Faculty of Mechanical Engineering, Technical University of Liberec, Studentská 1402/2, 461 17 Liberec, Czech Republic; jiri.safka@tul.cz (J.Š.); jiri.sobotka@tul.cz (J.S.); 2The Institute for Nanomaterials, Advanced Technologies and Innovation, Technical University of Liberec, Studentská 1402/2, 461 17 Liberec, Czech Republic; jakub.machacek@tul.cz (J.M.); michal.ackermann@tul.cz (M.A.)

**Keywords:** ceramics, fused filament fabrication, 3D printing, mechanical properties, relative density, shrinkage

## Abstract

This article deals with a comprehensive study of the processing and mechanical properties of the ceramic material Al${}_{2}$O${}_{3}$ on Fused Filament Fabrication technology (FFF). It describes the basic input analyses of the material, such as TGA, FTIR, and MVR. These analyses enabled the design and testing of process parameters for the 3D printing of parts. The article also presents the post-processes, including the technological parameters required to finalize parts made from this material, i.e., chemical debinding in acetone at elevated temperatures + thermal debinding and sintering. The microhardness was measured on the processed samples, depending on the density of the inner filling. The resulting hardness had an almost linear relationship between the percentage of filler (20–40–60–80–100%) and the resulting microhardness (1382–2428 HV10). Flexural strength was also measured on the test specimens with different degrees of internal filling (80–90–100%). However, inner filling do not affect the flexural strength (316.12–327.84–331.61 MPa). The relative density of the final parts was measured on a ZEISS METROTOM 1500 CT machine and reached 99.54%.

## 1. Introduction

Additive manufacturing (AM), also known as Three-Dimensional Printing (3D printing), is an innovative approach to the production of parts with complex geometry and internal structures. This innovative technology was invented and patented in 1984 by Charles Hull in a process known as stereolithography (SLA), the first commercial rapid prototyping technology from 3D Systems [1,2]. SLA technology was followed by subsequent developments such as Fused Deposition Modeling (FDM), Solid Ground Curing (SGC) from Cubital, and Laminated Object Manufacturing (LOM) from Helisys in 1991. Selective Laser Sintering (SLS) from DTM (now a part of 3D Systems) was developed in 1992 [3]. Fused Filament Fabrication (FFF), commercially known as FDM technology (trademarked by Stratasys, Ltd., Eden Prairie, MN, USA), usually uses thermoplastics. The basic principle of fabrication is to build a model layer by layer to achieve a 3D part. The raw material in the form of a filament is partially melted, extruded, and deposited onto the previously built model by a numerically controlled heated nozzle [4]. The most used thermoplastic polymers for FFF are acrylonitrile butadiene styrene (ABS), poly(lactic acid) (PLA), polypropylene (PP), thermoplastic polyurethane (TPU), and poly(ethylene terephthalate glycol) (PETG).

Thermoplastic filaments may be modified by the addition of fibers, powder, and other materials into the polymer matrix to form a composite, and increase their mechanical properties [5]. Fillers made from ceramic materials lead to enhanced mechanical or thermal properties and may be used for biological applications [6]. The AM of ceramics will complement and extend new possibilities of applications in the ceramics industry. Compared to conventional technologies such as Ceramic Injection Molding (CIM), AM offers new opportunities to manufacture ceramic components with a complex geometry without the need for expensive tooling molds, which leads to a reduction in production times, and consequently a reduction in costs and design flexibility [7]. The process of AM has a key advantage in the production of small quantities and customized parts. Functional prototype parts and tooling are manufactured directly from computer models [8]. CIM technology usually uses a binder system based on polyolefins and the waxes low density polyethylene (LDPE), poly(ethylene glycol) 6000 (PEG6000), paraffin wax (PW), carnauba wax (CW), Acrawax (AW), and stearic acid (SA), and the loading of the binder system varies from 14.5 to 15.8 wt% [9,10].

To obtain a final dense ceramic part, several steps are required. The 3D printing process represents only a procedure for the formation of material into the required shape. To achieve the final parts with the desired properties (such as mechanical properties, microstructure, geometry, etc.), it is necessary to take into account the chemical composition and ratio (size) of the ceramic particles, and the polymer matrix. Depending on the material composition and 3D printing technology used, post-processes such as debinding and sintering are required and have a fundamental influence on the final part [11].

Ceramic materials may be divided into two main groups: classic ceramics and advanced ceramics. Advanced ceramics are made by synthetic chemicals of high purity and organic binders are usually added to assist the shaping process. These specific ceramics are targeted to industrial applications that require high performance [12].

Alumina is one of the most commonly used and studied advanced ceramic materials due to the relative abundance and low cost of the material source, as well as the availability of the material in highly pure grades, which is used for material research [13]. Alumina is a known ceramic material for high heat resistance and high thermal conductivity, high tensile and compression strength, high electrical insulation, high corrosion resistance, chemical and physical stability, and biocompatibility. The material is hard and abrasive, and is resistant to thermal shock [14] Alumina parts are used in electrical and electronic applications [15], membrane [16] and filtration products [17], wear-resistant products such as sand blasting nozzles, seal faces, bearings and piston plungers [18], etc.

## 2. Materials and Methods

FFF is a process for the extrusion of thermoplastic material [19]. Generally, it is possible to use single material or composite material for FFF. A wide range of materials may be used as composite reinforcements (carbon, glass fibers, kevlars, ceramics, carbon nanotubes, wood, juta, palm, etc.) [20]. In our case, the composite system of the thermoplastics matrix is filled by ceramic powder (>45 vol%) and may be used to 3D print ceramic parts. In comparison with Powder Injection Molding technology, the Alumina powder loading is between 50 and 60 vol% [21]. Fundamental requirements for feedstock filaments are low viscosity, high strength, high strain, and high modulus [22]. To achieve dense ceramic parts, it is necessary to remove the binder system during the debinding process (chemical and thermal). A sintering step is required for the densification of ceramic particles. A schematic representation of the 3D printing and post-processes required to obtain the final ceramic part is given in Figure 1.

### 2.1. Material

Thermoplastic composite filament alumina used for the preparation of samples was fabricated by Zetamix (Nanoe, France). The alumina material and SEM pictures of the filament are shown in Figure 2. Alumina powder (Al${}_{2}$O${}_{3}$) with a ceramic particles size < 1.0 µm and a thermoplastic binder material were processed into a filament with a diameter 1.75 mm [23]. The filament is suitable for a technology FFF. The volume proportion of the polyolefin based binder system is 48 vol% and the alumina proportion is 52 vol%. Converted to a weight percentage concentration, the proportion of the binder system is 17 wt% and the alumina proportion is 83 wt% [23]. The porosity and mechanical properties of alumina fabricated by additive manufacturing and conventional technologies are compared in Table 1.

Thermogravimetric analysis (TGA) of the alumina material from Zetamix is shown in Figure 3, which provides information on the thermal decomposition of the polymer matrix. Decomposition begins at a temperature of 181 °C. The mean thermal decomposition temperature of the polymer matrix corresponds to 381 °C and the highest weight loss of the polymer being recorded at 385 °C. This temperature is an important point in thermal debinding process. Due to the previous chemical debinding, when a substantial part of the polymer matrix is removed, only the residual polymer is removed during the thermal debinding. A temperature of 526 °C indicates the end of thermal decomposition.

Furthermore, the melt volume rate (MVR) was also measured. At a temperature of 150 °C and a total sample load of 2.16 kg, the resulting MVR value was (159 ± 6) cm${}^{3}$/10 min. The measurement was performed in accordance with the international standard EN ISO 1133-1:2011. The measurement conditions were used for a polyethylene matrix, which was previously verified by FTIR spectrometry. Energy Dispersive X-ray Analysis (EDX), was used to verify the composition of the Al${}_{2}$O${}_{3}$ material on the sample before debinding and sintering.

### 2.2. Fabrication and Specimens

The specimens were printed on a FFF printer-Prusa i3 MK3S (Prusa Research, Czech Republic). Due to the fragility of the material, the pressure spring on the feed mechanism had to be replaced. The original spring damaged the filament, and this caused printing problems. A new spring with less pressure force partially grinded the material. The filament was not interrupted, and the material feeding process was continuous. Easy unwinding of the material from the spool was ensured using two ball bearings.

The z-axis adjustment plays a key role in obtaining a dense and smooth first layer of the part. A steel printing plate with a smooth polyetherimide (PEI) surface was used. The plate had to be properly cleaned after each printing to ensure smooth separation of the part.

The printing parameters recommended by the manufacturer (Zetamix by Nanoe) were optimized with regards to the achieved results and performed tests. The extrusion temperature was chosen based on a temperature test in the range of 115–190 °C in steps of 5 °C. At a low temperature of 110 °C, the material was not sufficiently melted and was not forced through the nozzle. At temperatures above 190 °C, the part warped due to low viscosity and high temperature.

The best results after the debindig and sintering process were obtained with an extrusion temperature of 150 °C. The whole test was repeated, and the selected temperature was confirmed to be optimal. The bed temperature of 25 °C ensured sufficient adhesion during the printing process, and smooth removal of the part from the build plate when the printing was completed. Layer height: 0.2 mm, speed: 30 mm/s, overlap: 40%, solid layers top/bottom: 2/2 (depending on a geometry of parts), perimeters: 2, infill pattern: rectilinear (Table 2).

A flexural strength test was performed according to the EN 843-1:2006 standard for special technical ceramics. The chosen specimen type A had the following dimensions: 2.5 × 2.0 × ≥25 mm (Figure 4).

### 2.3. Solvent Debinding

The debinding process has two steps: solvent and thermal. It is a crucial process for removing the a polyolefin-based binder system from the parts Figure 5. A Sonorex Digitec DT 510 H ultrasonic bath (BANDELIN Electronic GmbH & Co. KG, Berlin, Germany) was used for the solvent debinding process. The “green bodies” were impregnated in an acetone solvent bath. The time of solvent debinding varied depending on the size and geometry of the parts. The temperature was set in the range of 30–40 °C. Weight loss after the solvent debinding process is necessary to determine polymer matrix loss. The average weight loss value was 11%. In the event of insufficient binder removal, the specimens cracked. To avoid the cracks, it was necessary to leave the parts in an acetone bath for a sufficiently long time of 3–24 h (depending on the size of the parts).

### 2.4. Thermal Debinding and Sintering Process

The thermal debinding and sintering cycle was processed in a Clasic 1017S atmosphere furnace (CLASIC CZ s.r.o., Řevnice, Czech Republic). During the thermal debinding, the binder system is eliminated/removed by thermal energy. The sintering step is important for obtaining the final dense parts of pure alumina. Temperature as a function of time is show in Figure 6. The parts were re-weighed and measured to determine the percentage weight loss of all the binders after thermal elimination as well as geometry/dimension changes due to the sintering of grains. The maximum temperature of the debinding process reaches 510 °C, and the maximum sintering temperature is 1550 °C. These values, including the temperature ramp and endurance, were recommended by the manufacturer of the filament, and proved to be optimal.

After the debinding process, the part is called a “brown body”. The subsequent sintering process is required to achieve the final, densified part. The volume of the part is reduced. The sintering process may be divided into three categories, depending on the composition being fired, and in particular on the extent to which a liquid phase is formed during the heat treatment [32,33,34]. The mechanisms of sintering include solid state sintering, liquid-phase sintering, and viscous sintering. In this case, solid state sintering was used.

During this process, the green or brown body is heated to a temperature that is typically 0.5–0.8 of the melting temperature [35]. The sintering temperature of the Al${}_{2}$O${}_{3}$ material is usually between 1400 °C and 1650 °C, which is calculated from the melting point of Al${}_{2}$O${}_{3}$ of 2072 °C [36]. In solid-state sintering, the powder does not melt, and the composition and firing temperature are such that no liquid is formed. The particles are joined together and densification of powder is achieved. Diffusion of atoms is a mechanism forming and reshaping the powder. Energy reduction is achieved by elimination of the solid–gas interface and its replacement by a solid–solid interface, which causes reshaping [32,34]. Scanning electron microscope (SEM) images of a filament, a successfully sintered part, and an incorrectly sintered part due to low temperature, are shown in Figure 7.

## 3. Results and Discussion

### 3.1. Porosity

Porosity was measured on a ZEISS METROTOM 1500 CT machine (Carl Zeiss Industrial Metrology, Maple Grove, MN, USA). The printed parts (10 × 10 × 10 mm) with a 100% rectilinear filling were scanned after the sintering process. Individual CT scans of the specimen were processed using myVGL software (Volume Graphics GmbH, Heidelberg, Germany). On the basis of the data, 3D model of the specimen was reconstructed. The same software is capable of highlighting and evaluating porosity in the whole volume of the specimen. The largest porosity was in the area of perimeters (two perimeters were used for the printing). This may also be seen in Figure 8, up. The total porosity was 0.46% in the whole volume of the part. For comparison, the area of the perimeters was omitted, and the internal porosity was evaluated (Figure 8, down). The total porosity of the inner part was 0.28% (average porosity in the whole volume). To avoid porosity in the area of the perimeters, the extrusion width of the perimeters needed to be increased from 0.3 mm to 0.45 mm. The relative density was 99.54% (region of the perimeters included) and 99.72% (internal volume of the part without the perimeters region). In previous studies, Lithography-based Ceramics Manufacturing (LCM) technology had one of the highest relative densities of 99.3%, as reported by Schwentenwein and Homa Schwentenwein and Homa [25]. SLS technology reached a relative density of 95.66% and 88% according to Harris et al. [20] and Liu et al. [28], respectfully. In comparison with other AM technologies, the value of the relative density obtained in this experiment is one of the highest and it is comparable with conventional technologies.

### 3.2. Hardness

The hardness indicates the resistance to an applied load, which is indicated by a square-based pyramid [37]. The hardness of the specimens was measured on a Duramin-40 device (Struers Inc., Cleveland, OH, USA) according to the EN 843-1:2006 standard. The Vickers test method was chosen, which uses a diamond tip in the shape of a regular quadrilateral pyramid with a square base with a vertex angle between the opposite sides, alpha = 136 degrees. The penetrating body was pressed perpendicularly into the surface of the test specimen with a selected load of 10 kg (HV10) and a load time of 10 s. After unloading, the length of the impression diagonals was measured in two directions perpendicular to each other. The value of the resulting hardness was then automatically calculated by the software (the result is affected by the selected load force and the size of the impression diagonals). The influence of the internal filling of the test specimens on the resulting hardness was also evaluated. The inner fillings of 20–40–60–80–100% were chosen.

In order to perform the measurement, all the test specimens were closed with two solid layers on top. The height of each layer was 0.2 mm. It was not possible to measure microhardness in a semi-closed shell with zero filling. The diamond tip penetrated through both solid layers and the measurement was invalid. As may be seen in Table 3, the infill percentage had a significant effect on the hardness of the material, which increased almost linearly with increasing infill.

The hardness values were in the range of 1382 to 2428 HV10 (depending on the internal infill). The obtained values were recalculated to SI units using Equation (1).
(1)(GPa)=0.009807·HV

### 3.3. Flexural Strength

Flexural strength was determined according to the EN 843-1:2006 standard, and calculated by the standard Equation (2), where F [N] is the load (force) at the fracture point, *L* [mm] is the length of the support span, *b* [mm] is specimen width, *h* [mm] is specimen thickness. For each infill, five testing samples were used to obtain relevant data. The measured values of ranged from 316.12 to 331.61 MPa depending on the inner filling (Table 4). According to the data obtained, the internal filling in the range of 80–90–100% had no significant effect on increasing or decreasing the flexural strength. After the test, the fracture surfaces were analyzed by SEM (Figure 9). It is also possible to see porosity in the area of perimeters in the images of the fracture surfaces, which confirms the results from the CT. Therefore, it was necessary to adjust the process parameters of the printing. A lower porosity in this area may also lead to a higher flexural strength.

The flexural strength was approximately 30% higher compared to results obtained by SLS–255 MPa (4PB), reported by Li et al. [38], and 26% higher than conventionally manufactured alumina by powder injection molding with 264 MPa (3PB) reported by J. Y. Roh [31]. The results are 21% lower (427 MPa, 4PB) compared to the LCM technology reported by Schwentenwein and Homa Schwentenwein and Homa [25].
(2)$$\sigma_{f}=\frac{3Fl}{2bh^{2}}$$

### 3.4. Shrinkage and Geometry Deformation

The total shrinkage provided by the manufacturer of 21.3% should be obtained after the sintering step [23]. The values obtained by the experiment indicate the average shrinkage of 22.5%.

Partial shrinkage of the parts occurs after the debinding process, both chemical and thermal. There is also a significant weight loss after the chemical debinding process, with an average value of 12.45 wt%. During the thermal debinding process, the rest of the polymer is removed, and the part is extremely brittle after this step. However, the sintering process has the greatest influence on volume shrinkage. The high temperature compacts the ceramic particles and reduces the porosity. This is crucial for the production of high-precision parts with a defined geometry and dimension accuracy. Analysis of the shrinkage in all directions (x, y, z) is required. In the z-axis, there is a larger shrinkage due to the used technology and gravity. The orientation of parts during the printing process and thermal post-processing also plays a key role in axis shrinkage. The average weight loss is 22.6 wt%. The whole printing process for obtaining a dense ceramic part is shown in Figure 10. An example of the weight loss values after each process for parts with an internal gyroid structure is shown in Table 5.

## 4. Conclusions

This study demonstrated the fabrication of alumina material by FFF technology. The alumina material was analyzed by thermogravimetric analysis, FTIR spectrometry, EDX analysis, and MVR was measured. The volume proportion of the binder system was 48 vol% and the alumina particles was 52 vol%. Decomposition of the polymer matrix begin at 181 °C, point of greatest rate of change on the weight loss is at 381 °C. At temperature 526 °C thermal decomposition is finished. Due to these information, the thermal debinding process is under control. The printing parameters recommended by the filament manufacturer were modified to obtain sufficient results.

Chemical debinding renders the sample sensitive to cracking and delamination. To prevent damage to the samples, most of the binder must be removed in an acetone bath, and the process is controlled by the temperature and leaching time. The remaining polymer was removed during thermal debinding, which was followed by a sintering process. After the thermal debinding process the parts are extremely brittle.

The relative density was measured at 100% infill and reached 99.54%. The highest porosity was in the area of the perimeters, which were not completely sintered to each other. After forced elimination of this area, the relative density was 99.72%. The hardness was measured as a function of the infill percentage in the range of 20–100% (step 20%) and confirmed an almost linear increase in hardness with a higher percentage of infill. The maximum hardness was obtained with 100% infill and reached values of 2428 ± 209 HV10 (23.81 GPa). A three-point bending flexural test was performed on the specimens with internal infill ranges of 80–90–100%. However, the obtained results did not show any dependence on infill density. Flexural strength was in the range of 316.12–331.61 MPa. Shrinkage is a significant attribute of a composite system: polymer matrix and ceramic particles. After the sintering process, loose weight of the part was approximately 22.6 wt%. The obtained mechanical properties and relative density were comparable with those of conventionally manufactured parts.

## Figures and Tables

**Figure 1 polymers-14-00991-f001:**
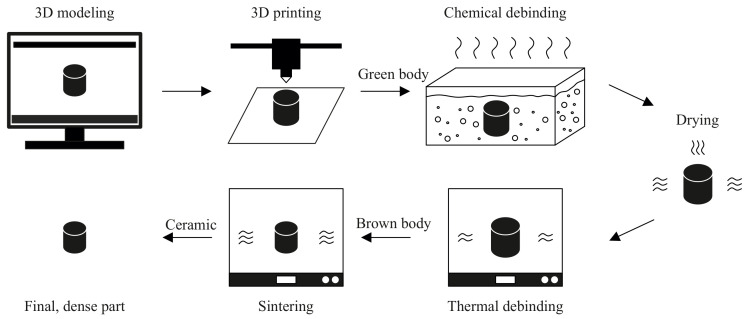
Schematic representation of 3D printing ceramics and post-processes.

**Figure 2 polymers-14-00991-f002:**
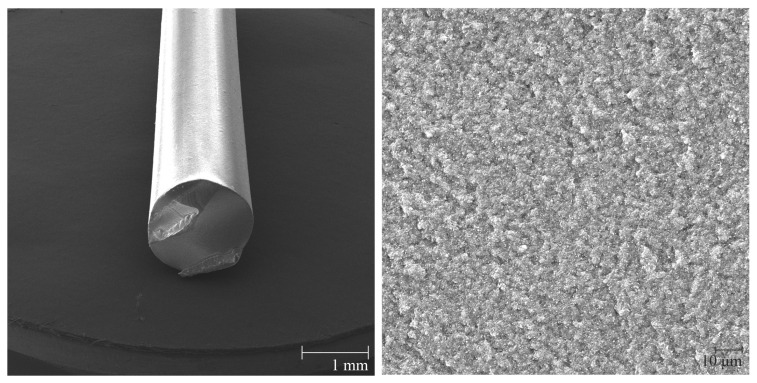
SEM view of the alumina filament and its cross-section.

**Figure 3 polymers-14-00991-f003:**
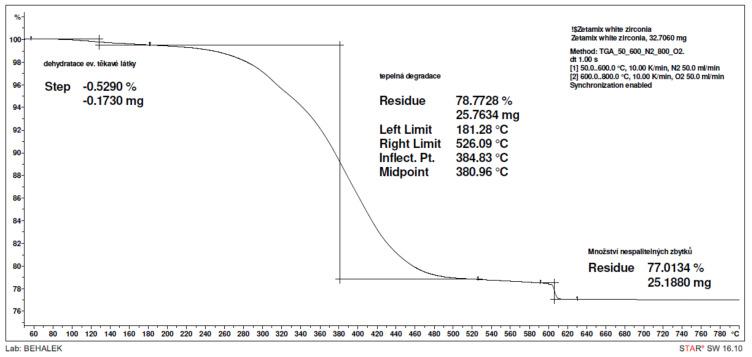
The thermogravimetric analysis of alumina material from Zetamix.

**Figure 4 polymers-14-00991-f004:**
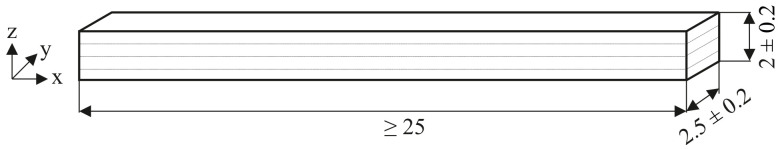
Flexural strength specimen type A [mm] with ilustration of layers direction.

**Figure 5 polymers-14-00991-f005:**
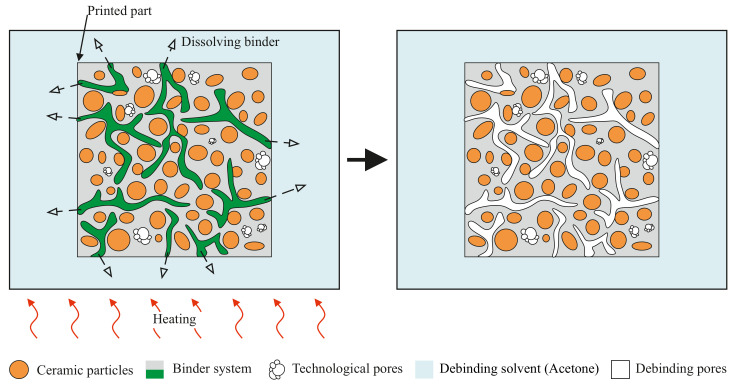
Illustration of solvent debinding process.

**Figure 6 polymers-14-00991-f006:**
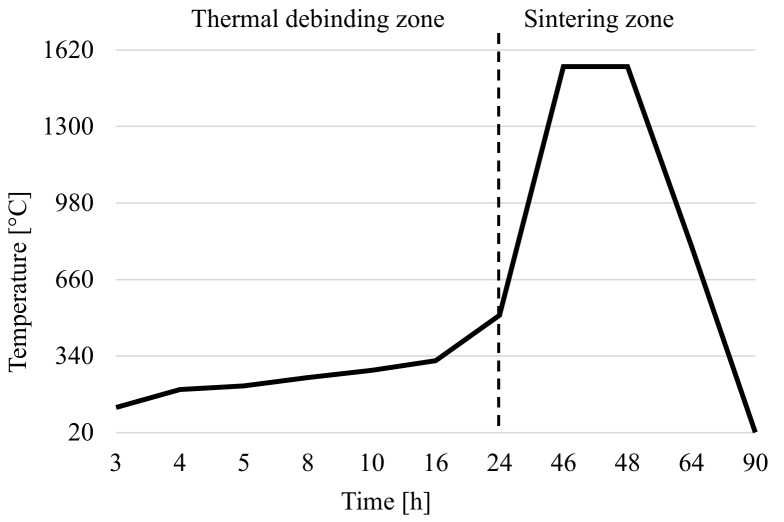
Thermal debinding and sintering cycle.

**Figure 7 polymers-14-00991-f007:**
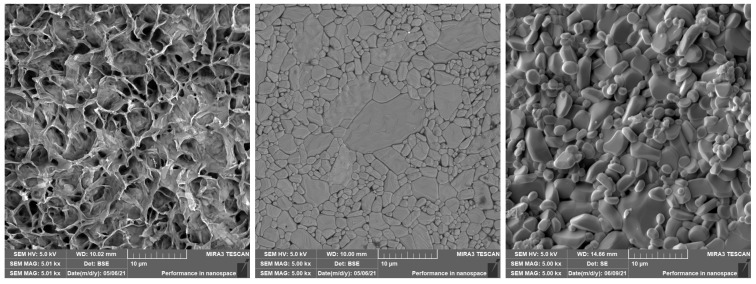
SEM images of: (**left**) the filament, (**middle**) a sintered part, (**right**) a part sintered unsuccessfully (sintering process with a low temperature).

**Figure 8 polymers-14-00991-f008:**
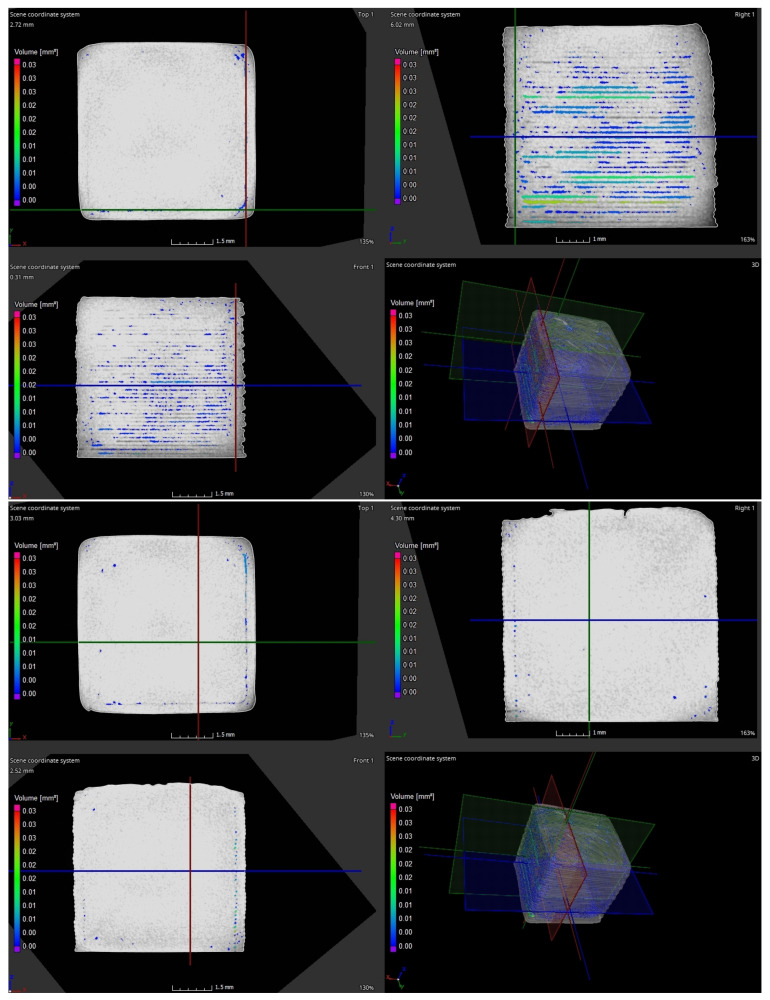
Data from CT ZEISS METROTOM 1500 of sample with 100% infill. Porosity in the area of perimeters (**up**). Porosity of the inner area of the part (**down**).

**Figure 9 polymers-14-00991-f009:**
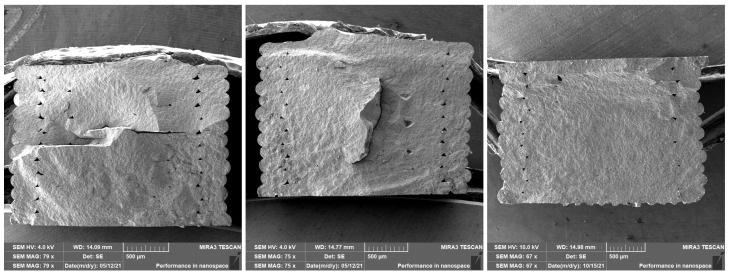
Fracture surface analysis with internall infill, left: 80%, 90%, and 100% infill.

**Figure 10 polymers-14-00991-f010:**
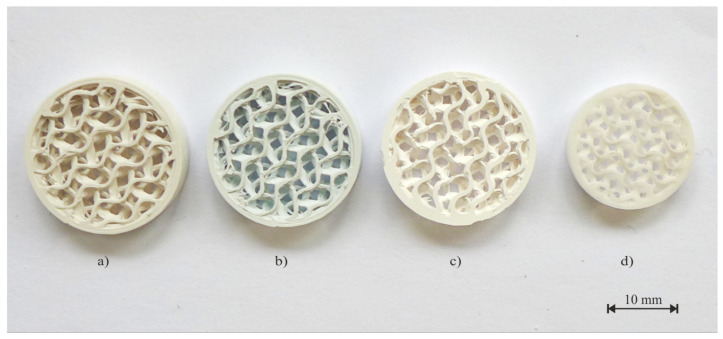
Part with an internal gyroid structure: (**a**) Printing on FFF technology, (**b**) Chemical debinding, (**c**) Thermal debinding, (**d**) Sintering process.

**Table 1 polymers-14-00991-t001:** Alumina material fabricated by AM and conventional technologies. Comaprison of porosity and mechanical properties reached in this study with results of other research teams.

Manufacturer	Technology	Flexural Strength [MPa]	Hardness	Relative Density [%]	Citation
Sumitomo Chemical Co., Ltd., Tokyo, Japan	Fused Filament Fabrication	200–300	-	80–89	[24]
LithaLox HP 500, Lithoz, VIenna, Austria	Lithography-based Ceramics Manufacturing	427 (4PB)	-	99.3	[25]
Kemaus, Australia and A32, Japan	Photosensitive Binder Jetting	1.04	-	39.49	[26]
Taimicron—KRAHN, Hamburg, Germany	Selective Laser Sintering	-	-	95.66	[27]
Almatis Inc., Ludwigshafen, Germany	Selective Laser Sintering	255 ± 17 (4PB)	-	88	[28]
Fenghe Ceramias Co., Ltd., Shanghai, China	Tape Casting	-	15.91 ± 0.15 GPa	98.1	[29]
Alcan Chemicals, Stamford, CT, USA	Slip Casting Technique	-	1679 HV30 (16.46 GPa)	98.04	[30]
Sumitomo Chemical, Tokyo, Japan	Powder Injection Molding	264 (3PB)	1903 HV200 (18.66 GPa)	99.5	[31]
Zetamix, Nanoe, France—Datasheet	Fused Filament Fabrication	150–300	19 GPa	98–99	[23]
Zetamix, Nanoe, France—Results	Fused Filament Fabrication	316.12–331.61 (3PB)	13.54–23.81 GPa	99.54 (99.72)	

**Table 2 polymers-14-00991-t002:** Printing parameters for alumina material on FFF technology.

Printing temperature	150 °C
Bed temperature	25 °C
Layer height	0.2 mm
Speed	30 mm/s
Solid layers Top/Bottom	2/2
Retraction	off
Overlap	40%
Perimeters	2
Infill pattern	rectilinear

**Table 3 polymers-14-00991-t003:** Hardness of alumina depending on the infill percentage.

Infill [%]	Hardness HV10	Hardness [GPa]
20	1382 ± 191	13.54
40	1521 ± 237	14.91
60	1758 ± 188	17.23
80	2178 ± 340	21.34
100	2428 ± 209	23.81

**Table 4 polymers-14-00991-t004:** Flexural strength depending on the internal infill.

Infill [%]	Flexural Strength [MPa]
80	316.12 ± 58.77
90	327.84 ± 26.21
100	331.61 ± 53.45

**Table 5 polymers-14-00991-t005:** Part with an internal gyroid structure: (a) Printing on FFF technology, (b) Chemical debinding, (c) Thermal debinding, (d) Sintering process.

	Units	3D Printing	Chemical Debinding	Thermal Debinding	Sintering
Weight	g	1.33	1.15	1.05	1.03
Processing shrinkage	wt%	-	13.53	7.55	1.55
Total shrinkage	wt%	-	13.53	21.05	22.56

## Data Availability

Not applicable.

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
