# Peer review of "Alumina Manufactured by Fused Filament Fabrication: A Comprehensive Study of Mechanical Properties and Porosity"

_polymers, 2022, doi:10.3390/polym14050991_

Round 1
Reviewer 1 Report
- The authors should include results in the abstract.
- Line 28: poly(lactic acid)
- Line 29: poly(ethylene terephthalate glycol)
- Line 43: poly(ethylene glycol)
- Section 2.1 What was the polymer matrix?
- Figure 3 right: was the SEM image the crosssection of the filament or the surface?
- Figure 8: The authors need to update the figure caption.
- Section 3.1: Was the porosity calculation based on one slice of the CT image or was it the average result of all the CT slices in certain regions?
- Line 186: What do the authors mean 'The relative density was 99.54% (99.72%)'? Do the authors mean the relative density was increased from 99.54% to 99.72% with the increase of extrusion width of the perimeters?
- Line 223 and Table 4: The flexural strengths of the parts with different infill% are not statistical significant different.
- Line 230: What is b? h? l? Explanation of each symbol is needed.
Author Response
Dear Reviewer,
Thank you very much for reading and evaluating of our article. We are glad for your suggestions which lead to increase the quality of our work. Our team of authors went through all of your points and made following changes in the manuscript (please see the attachment).

Reviewer 2 Report
The paper is about the investigation of the processing and mechanical characteristics of the ceramic material Al2O3 using Fused Filament Fabrication technology. TGA, FTIR, and MVR were among the methods used in the study. The current work is excellently written, and I urge that it be published following a slight technical revision:
- The document contains several unneeded abbreviations.
- Change [12] to [12] and [21] to [21].
- It is necessary to define the parameters in equation (2).
- The TGA analysis and Table 1 should be deleted from the section findings and discussion.
Author Response
Dear Reviewer,
Thank you very much for reading and evaluating of our article. We are glad for your kind review and suggestions which lead to increase the quality of our work. Our team of authors went through all of your points and made following changes in the manuscript (please see the attachment).
Best Regards,
Veronika Truxova
